# Application of Natural Deep Eutectic Solvents for Extraction of Bioactive Components from *Rhodiola rosea* (L.)

**DOI:** 10.3390/molecules28020912

**Published:** 2023-01-16

**Authors:** Nikita Tsvetov, Oksana Paukshta, Nadezhda Fokina, Natalia Volodina, Artemiy Samarov

**Affiliations:** 1Tananaev Institute of Chemistry and Technology of Rare Elements and Mineral Raw Materials—Subdivision of the Federal Research Centre «Kola Science Centre of the Russian Academy of Sciences», Akademgorodok 26a, 184209 Apatity, Russia; 2Chemistry Department, Murmansk State Technical University, Sportivnaja Str. 13, 183010 Murmansk, Russia; 3Institute of North Industrial Ecology Problems—Subdivision of the Federal Research Centre «Kola Science Centre of the Russian Academy of Sciences», Akademgorodok 14a, 184209 Apatity, Russia; 4Department of Chemical Thermodynamics and Kinetics, Institute of Chemistry, Saint Petersburg State University, Universitetskiy Prosp. 26, 198504 St. Petersburg, Russia

**Keywords:** natural deep eutectic solvents, *Rhodiola rosea* (L.), ultrasound-assisted extraction, antibacterial activity

## Abstract

*Rhodiola rosea* (L.) is a valuable source of nutrients. Nutrients have adaptogenic, immunostimulating, nootropic, anti-inflammatory and anti-cancer properties. Natural deep eutectic solvents (NADES) consisting of choline chloride and malonic, malic, tartaric or citric acids have been first used to extract biologically active substances from *R. rosea*. The total content of polyphenols has been determined by the Folin–Ciocalteu method for all extracts. Antioxidant activity has been determined by the phosphomolybdate method, and antiradical activity has been determined by the 2,2-diphenyl-1-picrylhydrazyl (DPPH) method. Rosavin concentration has been determined by high-performance liquid chromatography (HPLC). Extraction kinetics has been evaluated regarding the effectiveness of NADES with each other and with reference solvents (water and 50% ethanol) has been made. Extraction conditions have been optimized according to the Box–Behnken design of the experiment. The optimal parameters of the extraction process have been established. The antibacterial activity of NADES-based extracts against bacterial cultures of *Micrococcus luteus*, *Pseudomonas fluorescens*, and *Bacillus subtilis* has been studied.

## 1. Introduction

*Rhodiola rosea* (L.) (syn. *Sedum rhodiola*-DC. *Sedum rosea*-(L.) Scop) is a perennial herbaceous plant of the Crassulaceae family, widely distributed in the Arctic regions of Eurasia, North America, and Mongolia. It grows on rocky mountain slopes, in periodically flooded areas, and along the coasts of the northern seas. This plant is used in both folk and official medicine because its extracts have adaptogenic, immunostimulating, and nootropic properties. The main amount of bioactive components of the plant accumulates in rhizomes. The main components exhibiting biological activity are glycosides of tyrosol (salidroside) and cinnamic alcohol (rosin, rosavin, rosarin), and monoterpene alcohols and their glycosides (geraniol, rosiridol, rhodiolosid, etc.) [1]. Volatile substances contained in *R. rosea* are n-decanol, geraniol, 1,4-p-menthadien-7-ol, benzyl alcohol, and phenylethyl alcohol [2,3].

The traditional use of *Rhodiola rosea* has been known since Viking times. This plant was described by Linnaeus, who recommended it for the treatment of ‘hysteria’ and a number of other ailments. *Rhodiola rosea* is used to improve performance and as a stimulant. Its positive effect on the nervous system is known. In Russia, the liquid extract of *Rhodiola rosea* is used as an adaptogen [1]. Some substances contained in *R. rosea* exhibit anti-inflammatory and anti-cancer activity and reduce the risk of developing neurodegenerative diseases [4]. Salidroside is considered to be a valuable commercial product with high biological activity [5]. *R. rosea* extracts exhibit high antioxidant and antimicrobial activity. For example, the properties of dry ethanol extract from rhizomes obtained at 50 °C in 60% aqueous ethanol (*v*/*v*) with a ratio of plant material to solvent of 1:10 (*m*/*v*) were studied in work [2]. The tests were carried out using strains of *Bacillus cereus*, *B. cereus*, *B. cereus*, *Bacillus subtilis*, *Staphylococcus aureus*, *Staphylococcus epidermidis*, *Listeria monocytogenes*, *S. aureus Enterobacter aerogenes*, *Escherichia coli*, *Klebsiella pneumoniae*, *Proteus mirabilis*, *Salmonella enterica subsp. enterica serovar Enteritidis*, *Pseudomonas aeruginosa*, *S. enterica subsp. enterica serovar Enteritidis*, *Salmonella enterica subsp. enterica serovar Typhimurium*, *Yersinia enterocolitica*, and *Shigella sonnei*. The high antibacterial activity of the extracts was shown. Adaptogenic, antiviral and antibacterial effects of extracts against *Escherichia coli* and *Staphylococcus aureus* strains were demonstrated in experiments involving marathon runners. However, no significant inhibitory effect was found [6]. At the same time, the work [7] significant antibacterial effect of aqueous and water-ethanol extracts of *Rhodiola rosea* against *Pseudomonas aeruginosa* infection in mice showed. In Ref. [8], a strong antibacterial activity of ethyl acetate extract of *Rhodiloa crenulata* was demonstrated.

The occurrence of this plant is rare in the wild nature of some regions. More frequently, the plant is cultivated; however, active compounds are accumulated in it for several years [9,10], therefore, the development of new, more efficient ways to extract bioactive components from this plant material is a relevant challenge. Usually, the biologically active substances are extracted with methanol or ethanol [11,12,13]; the extraction is combined with elevated pressure or microwaving [14,15]. Supercritical extraction is used in some cases [16].

Deep eutectic solvents (DES) and natural deep eutectic solvents (NADES) [17,18] are promising for the extraction of biologically active compounds [19]. They extract quite well various glycosides and compounds having OH groups such as flavonoids and polyphenolic acids [20]. Therefore, it can be assumed that the main biologically active compounds of *R. rosea* can be successfully extracted using NADES. The literature devoted to the use of NADES for the extraction of secondary metabolites from *R. rosea* is scarce [21,22,23]. At the same time, a suitable NADES composition capable of extracting rosavin and salidroside in larger amounts than traditional solvents was not selected in some works [23]. The goal of our research is to produce comprehensive research on the extraction of biologically active substances from *R. rosea* using NADES. These solutions consist of choline chloride and malonic, malic, tartaric or citric acids.

## 2. Results and Discussion

The change in the content of rosavin in the extracts increases in the first 40–60 min of extraction (Figure 1). A drop in concentration is observed after this time. The drop in concentration may be due to the oxidation of rosavin in the air and the effect of ultrasound. A similar picture is observed for TPC, TAC and DPPH. At the same time, the TPC indicator reaches equilibrium in the first 20 min of extraction in the case of water and ethanol, and the extraction of polyphenols in NADES is slower. A drop in the content of polyphenols is observed in most solvents after 2 h of extraction. The antioxidant activity of the extracts in the case of water and ethanol monotonically decreases after 20–40 min of extraction, and a strong drop is not observed in the case of NADES. It can be concluded that the complex of substances responsible for TAC is more resistant to oxidation in the NADES medium than the reference solvents. Antiradical activity abruptly changes in the first 20–40 min of extraction and then reaches a plateau. As with TAC, a monotonic drop in DPPH for ethanol extracts is observed. The complex shape of the kinetic curves indicates the presence of several processes. They are presumably associated with both the diffusion of target substances into the solvent and their destruction.

The comparative efficiency of different solvents was determined by the parameters TPC, TAC and DPPH of the extracts obtained after ultrasonic treatment at 45 °C for 60 min. NADES with malonic and tartaric acids (Figure 2a) showed the highest efficiency in extracting polyphenols, outperformed reference solvents by 10–20 mg GAE/g and reached a value of 160 mg GAE/g. The antioxidant activity of all NADES-based extracts was higher than for reference solvents by 10–20 mg AAE/g and ranged from 55–65 mg AAE/g (Figure 2b). The highest antioxidant activity was observed for mixtures of choline chloride with malonic, malic and citric acids among extracts with NADES. The antiradical activity of extracts based on NADES turned out to be lower than that of reference extractants and amounted to 55–60%. Only NADES with malonic acid showed the highest degree of DPPH inhibition, about 80% (Figure 2c).

Evaluation of the statistical significance of the difference in the results obtained using Tukey’s HSD test (*p* < 0.05) showed: differences in the TPC parameter are significant for all pairs of extractants, except for ethanol and citric acid; for the TAC parameter, it is significant for all pairs of extractants, except for NADES with malonic and malic, malonic and citric acids; for DPPH inhibition, it is significant for all pairs of solvents, except for the pair of ethanol–water, ethanol–NADES with tartaric acid and NADES with malic and citric acids (Table 1).

In general, in terms of the efficiency of rosavin extraction, NADES are similar to an aqueous-ethanol mixture, which was also shown in [23] in the example of lactic acid-based NADES with fructose and glucose. In the work [23] the extraction yield of rosavine for a 40% ethanol-water mixture was about 12 mg/g, for NADES (lactic acid:fructose:H_2_O (5:1:5)—about 10 mg/g.

NADES choline chloride + malonic acid was chosen for further experiments to optimize extraction conditions. The results of experiments to optimize extraction conditions in accordance with the BBD were approximated by polynomials of the second degree:RY = 696.905 − 21.513 A − 3.658 B + 197.644 C + 4.612 AB + 10.188 AC + 46.674 BC − 57.706 A^2^ + 87.517 B^2^ + 75.183 C^2^
TPC = 238.580 − 4.825 A + 24.438 B + 5.813 C 24.175 AB − 2.175 AC + 10.900 BC − 58.615 A^2^ − 12.140 B^2^ − 35.840 C^2^
TAC = 30.120 − 0.713 A + 0.738 B − 0.525 C + 0.775 AB + 1.950 AC + 0.750 BC − 6.548 A^2^ − 0.998 B^2^ − 1.223 C^2^
DPPH = 37.800 − 0.325 A − 1.263 B − 0.438 C + 2.350 AB − 1.650 AC − 2.225 BC − 4.488 A^2^ + 0.338 B^2^ + 0.188 C^2^
where RY is rosavin yield, A is temperature, B is time, and C is water content in NADES.

The dependence of RY, TPC, TAC and DPPH on temperature, time of extraction and water content in NADES was illustrated on the response surface plots generated by these equations (Figure 3, Figure 4, Figure 5 and Figure 6). The correlation coefficient R^2^ was 0.864 for RY, for TPC was 0.840, for TAC it was 0.713, and for DPPH it was 0.897. This indicates a fairly good approximation of the experimental data.

The results of ANOVA (Table 2) showed that the significant parameters of the model are AB, BC, and A^2^ for RY. The significant model parameters are B, A^2^, and C^2^ for TPC. Significant is the A^2^ parameter for TAC, and the significant model parameters are AB, BC, and A^2^ for DPPH, as well as for RY.

The optimal extraction parameters were calculated: the temperature was 46 °C, the time was 60 min, and the addition of water was 43 wt%. The concentration of rosavin under these conditions was 963.4 µg/mL, TPC was 248.2 mg GAE/g, TAC was 29.6 mg AAE/g, and DPPH was 35.3%.

The study of antibacterial activity was carried out in several stages.

It was found that the extracts in which water was used as a solvent in the experiment with discs and bacterial cultures of *Micrococcus luteus*, *Pseudomonas fluorescens*, and *Bacillus subtilis* did not show bactericidal properties at the first stage (Figure 7). Moreover, the rest of the extracts at the initial concentration and further diluted by 2 and 10 times inhibited bactericidal growth (Figure 8).

Numerical indicators of the bactericidal effect of extracts and solvents used in different concentrations for the studied bacterial strains were obtained at the second stage of research. The obtained results showed that all the studied extracts, except for the aqueous extract, exhibit a sufficiently high bactericidal activity even at a concentration of 0.5% and a sufficiently high initial number of bacteria (Table 3). This is true for both gram-positive and gram-negative bacteria that do not form spores. The bacterial strain *Micrococcus* sp. showed the highest susceptibility to all extracts. Its complete cell inhibition occurs when a 0.5% solution of all studied extracts is used. *Escherichia coli* cells showed the lowest susceptibility. These data are consistent with the data obtained in Ref. [6], where *Escherichia coli* were resistant to Rhodiola rosea extracts. Their complete growth inhibition occurs when 2% solutions of MA and Mal solvents are used. Experiments were carried out using the spore-forming bacteria *Bacillus brevis*. These experiments showed a rather low bactericidal activity of all studied extracts at a concentration of 50%; the number of bacteria decreased by 2–20 times (Table 4).

When an aqueous extract of *Rhodiola rosea* was used, the bactericidal effect was less pronounced for all groups of bacteria studied (Table 5). For example, the abundance of the most sensitive bacterial strain *Micrococcus* sp. decreased by a factor of 235 when 50% aqueous extract was used. On the contrary, the abundance of the strain *Micrococcus* sp. decreased only 1.5 times when 2% of the extract was used.

To compare the bactericidal effect of *Rhodiola rosea* extract and the studied extractants, an experiment was set up using a bacterial strain of *Escherichia coli*. The obtained results showed that the efficiency of the extracts was 11, 30 and 57 times higher than that of Tar, MA and CA (Table 6). The difference could not be detected in the case of using the extract and the Mal solvent since the used concentration led to the suppression of bacterial growth.

## 3. Materials and Methods

### 3.1. Materials

Following reagents were used in this study: malonic, malic, tartaric and citric acids (>99%, Vekton, St. Petersburg, Russia), choline chloride (99%, Rongsheng Biotech, Shanghai, China), salidroside and rosavine analytical standard (Sigma-Aldrich, Burlington, MA, USA), ammonium molybdate, potassium dihydrogenphosphate, aluminium chloride (>99%, Vekton, Yekaterinburg, Russia), Folin–Ciocalteu reagent (2M, Sigma-Aldrich, USA), 2,2-diphenyl-1-picrylhydrazyl (99%, Sigma-Aldrich, St. Louis, Missouri, USA), concentrated sulfuric acid (>94%, Nevareactiv, St. Petersburg, Russia), rutin (≥94%, Sigma-Aldrich, USA), ascorbic acid (>99.7 %, Hugestone, Nanjing, China), gallic acid (98% Sigma-Aldrich), ethanol (96%, RFK Company, Moscow, Russia), and deionized water obtained with water purification system “Millipore Element” (Millipore, Burlington, MA, USA).

Commercially available samples of rhizomes of R. rosea collected in the Altai region were used. Plant material was powdered with a blade grinder, sieved with laboratory sieves (0.1–0.5 mm) and dried at 40 °C to a constant mass. Eutectic solvents were prepared by mixing the components followed by heating to 60–70 °C in a choline chloride:acid molar ratio of 1:1 for malonic, malic, and citric acids and 2:1 for tartaric acid. As a result, transparent colorless liquids were obtained: choline chloride, malonic acid 1:1 (MA), choline chloride, malic acid 1:1 (Mal), choline chloride, citric acid 1:1 (CA) and choline chloride, tartaric acid 2:1 (tar). Sample preparation and characterization are described in detail in Ref. [24].

### 3.2. Methods

#### 3.2.1. Extraction

Samples of 0.05 g of plant material were mixed in Eppendorf tubes with 0.5 mL of solvent. The tubes were placed in an ultrasonic bath VBS-3DP (Vilitech, Moscow, Russia) filled with 3 L of water. Sonicated was performed with power 120 W and frequency 40 kHz for a selected time at a certain temperature. Thereafter, the tubes were centrifuged at 4 krpm for 5 min and the supernatant was discarded. To study the extraction kinetics, the temperature was set to 45 °C, and the extraction time was 20, 40, 60, 120, and 180 min. A comparison of the efficiency of solvents was made according to the results of the analysis of extracts obtained after 60 min of extraction at 45 °C. To optimize the extraction conditions, the Box–Behnken design of the experiment (BBD) was chosen. The temperature was 30, 45 and 60 °C, the extraction time was 20, 40, and 60 min, and the addition of water to NADES was 10, 30 and 50 wt%.

Water and a mixture of ethanol and water 90% *v*/*v* were used as reference solvents.

#### 3.2.2. Chemical Analysis

The chemical analysis of the extracts was carried out in accordance with the methods described in the works [24,25]. Briefly, total polyphenols content (TPC) was evaluated using the Folin–Ciocalteu method. The extract was diluted 100 times in ethanol for the Folin–Ciocalteu method. Total polyphenols content values are expressed in mg of gallic acid equivalent (GAE) per 1 mL of extract.

#### 3.2.3. In-Vitro Biological Activities Analysis

Total antioxidant activity (TAC) was evaluated by the phosphomolybdate method. The antiradical activity was evaluated by the DPPH method. Both procedures were described in detail in works [24,25]. The extract was diluted 400 times in ethanol for the DPPH method. To compare DPPH parameters for extracts based on different extractants, a 100-fold dilution was used. Total antioxidant activity values are expressed in mg of ascorbic acid equivalent (AAE) per 1 mL of extract.

#### 3.2.4. Analysis of Rosavin by HPLC

Analysis was done using an LC-20 prominence liquid chromatograph system (Shimadzu, Kyoto, Japan) with a diode-array detector SPD-M20A. A chromatographic column Luna C18 (2) (Phenomenex, 250 mm × 4.6 mm, 5 μm) was used. The column temperature was 40 °C. The gradient program for water with 0.1% *v*/*v* formic acid (A) and methanol (B) with a flow rate of 1 mL/min was as follows: as mobile phases, the gradient elution mode was used: 0–5 min, 5% C; 5–37 min, 5–90% B; 37–45 min, 90% B; 45–50 min, 90–5% B; 50–55 min, 5% B. The injection volume was 20 µL. The calibration curve was constructed using rosavin solutions at concentrations of 100, 50, 25, 12.5 and 6.25 μg/mL (R^2^ = 0.9977). The extracts were diluted 100-fold in methanol and filtered with a 0.45 μm syringe filter before analysis.

#### 3.2.5. Antibacterial Activity

Gram-positive and gram-negative bacteria, often found in indoor air, were used as study objects. Micrococcus luteus among gram-positive bacteria was used. Pseudomonas fluorescens and Escherichia coli among the gram-negative were used. The spore-forming bacterium Bacillus subtilis also served as an object. Micrococcus luteus (VKM Ac-2228), Pseudomonas fluorescens (VKM B-526), Bacillus subtilis (VKM B-501T). The strain Escherichia coli has not yet been deposited and is contained in the Museum of bacteria and fungi of Kola peninsula INEP Herbarium of Institute of the Industrial Ecology Problems of the North of the Kola Science Center of the Russian Academy of Sciences.

Experience with disks was set at the first stage. A bacterial suspension was prepared with a cell number of 108 cells/mL. Checking the bacteria number in the suspension was determined by the method of inoculation. The bacterial suspension (0.1 mL) was sown in Petri dishes on a solid medium of meat-peptone agar (MPA). A disc impregnated with extracts at various concentrations was placed in the center of the Petri dish after seeding. Bacteria were cultivated on Petri dishes for 3 days at 27 °C. The result was evaluated by the presence of a zone in which bacterial growth was absent. At the second stage, extracts and solvents at different concentrations were added to the bacterial suspension, and inoculation was carried out on a solid nutrient medium after 10 min, and the change in the number was monitored at the second stage.

#### 3.2.6. Statistical Analysis

All measurements were made three times for each analysis. Statistically significant differences were compared at *p* ≤ 0.05 using ANOVA and Tukey’s HSD (honestly significant difference) tests in MS Excel 2010 (Microsoft, Redmond, DC, USA). ANOVA test and Response Surface Methodology (RSM) were used for the data processing of BBD to optimize the ultrasound-assisted extraction conditions. DesignExpert 11 (Stat-Ease, Minneapolis, MN, USA) software was used for the ANOVA and RSM optimization.

## 4. Conclusions

In this work, for the first time, the ultrasonic extraction of biologically active substances from *Rhodiola rosea* using four NADES compositions of choline chloride + malonic, malic, tartaric or citric acid was comprehensively studied. HPLC showed that the main extraction process is completed within an hour. Optimization of extraction conditions using the Box–Behnken design of the experiment was done. Optimal extraction parameters: the temperature was 46 °C, the time was 60 min, and the addition of water was 43 wt%. An antibacterial effect of obtained extracts was tested using *Micrococcus luteus*, *Pseudomonas fluorescens*, and *Bacillus subtilis.* A high antibacterial effect of both NADES and extracts based on them was shown. At the same time, the antibacterial effect of the extracts exceeded by about 10–60 times that of the solvents.

Generally, it may be concluded NADESs based on choline chloride and some organic acids are a good alternative to ethanol for the extraction of salidroside. In addition, the antibacterial activity of both solvents and extracts can contribute to prolonging the shelf life of extracts and outlines ways to further use such extracts.

The obtained data can be used for further development of technologies for extracting bioactive components from *Rhodiola rosea* using NADES.

## Figures and Tables

**Figure 1 molecules-28-00912-f001:**
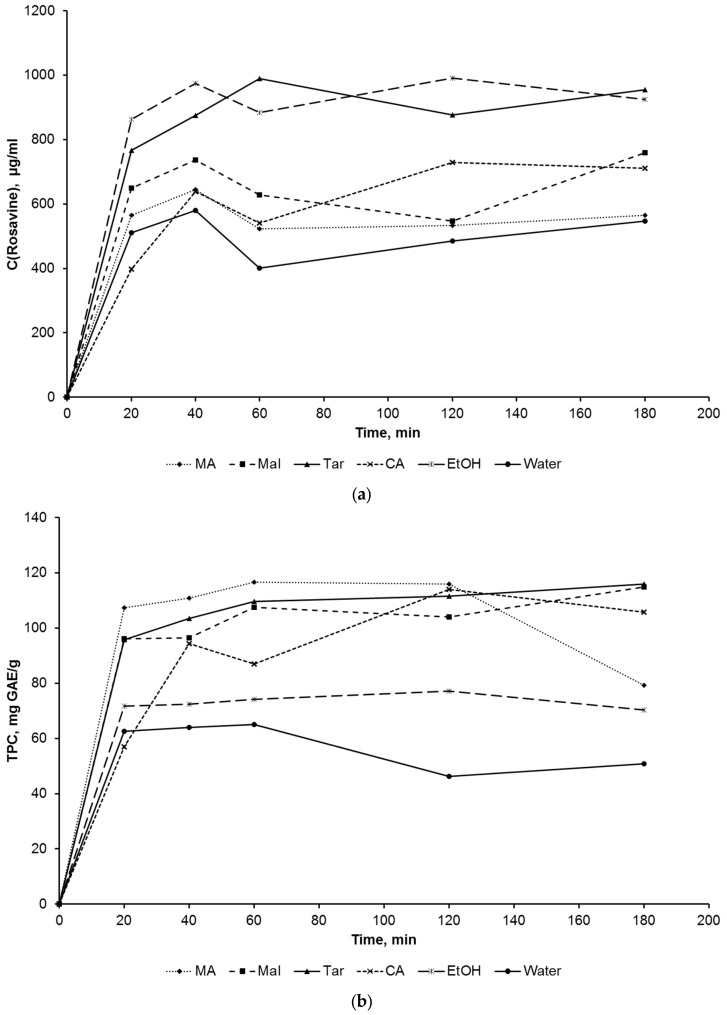
Time dependence of rosavin (**a**), TPC (**b**), TAC (**c**), and DPPH (**d**) concentrations for NADES-based choline chloride and malonic (MA), malic (Mal), tartaric (Tar), and citric (CA) acids, 90% *v*/*v* ethanol (EtOH) and water.

**Figure 2 molecules-28-00912-f002:**
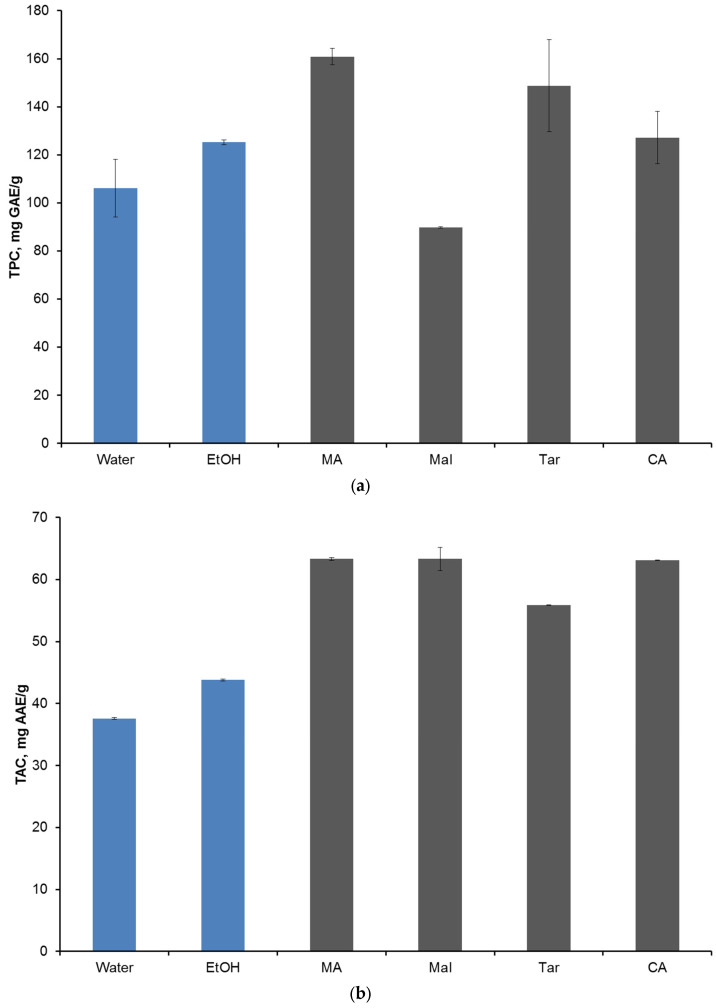
Comparison of the efficiency of various solvents in terms of TPC (**a**), TAC (**b**) and DPPH (**c**) at 45 °C for 60 min. Blue color is used for reference solvents.

**Figure 3 molecules-28-00912-f003:**
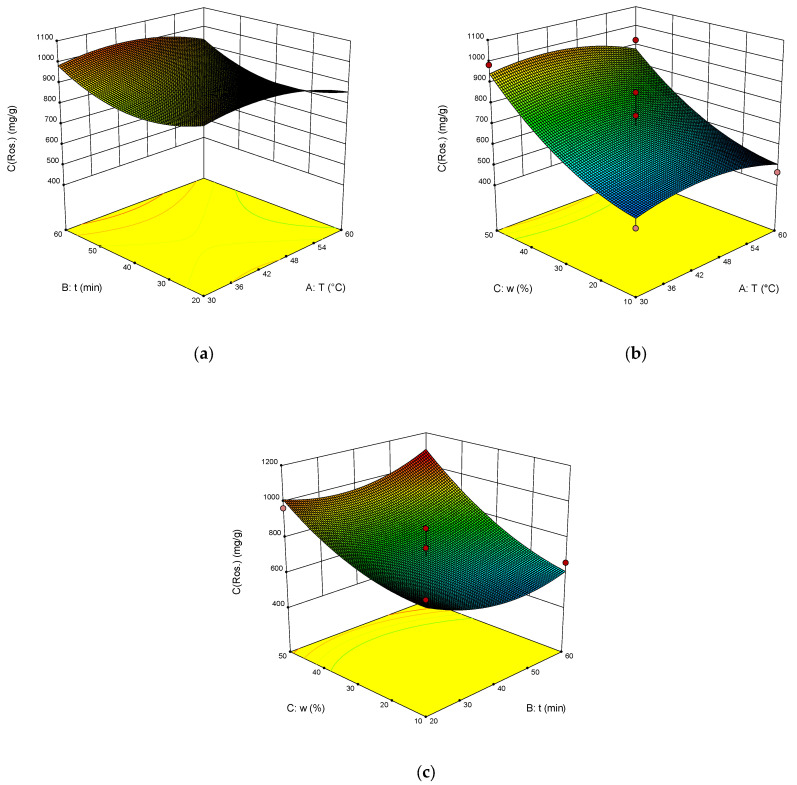
The response surface (**a**–**c**) shows the effect of extraction temperature (A), extraction time (B) and water content (C) on the extraction yield of rosavin.

**Figure 4 molecules-28-00912-f004:**
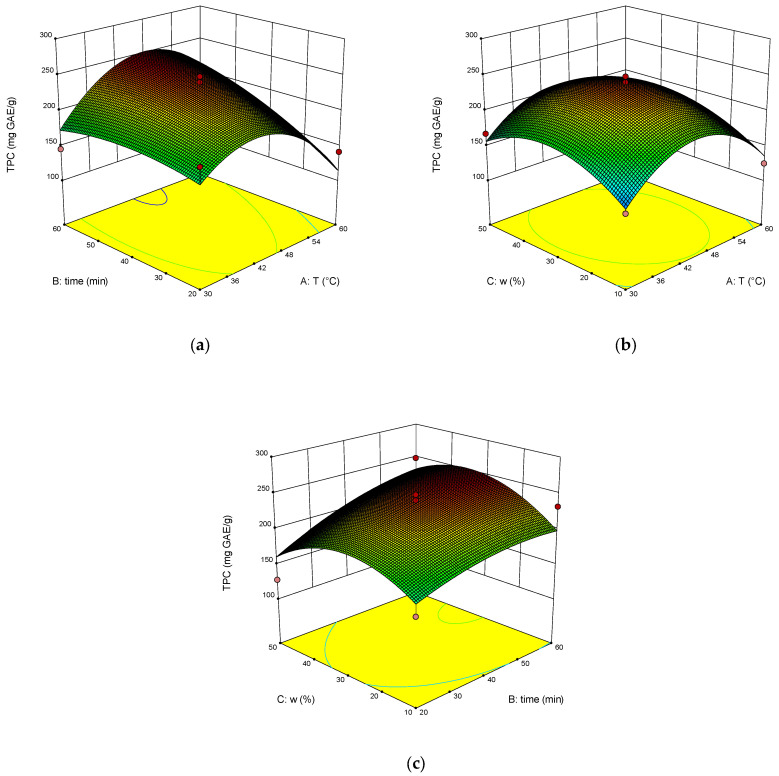
Response surface (**a**–**c**) shows the extraction temperature (A), extraction time (B) and water content (C) effect on the extraction yield of TPC.

**Figure 5 molecules-28-00912-f005:**
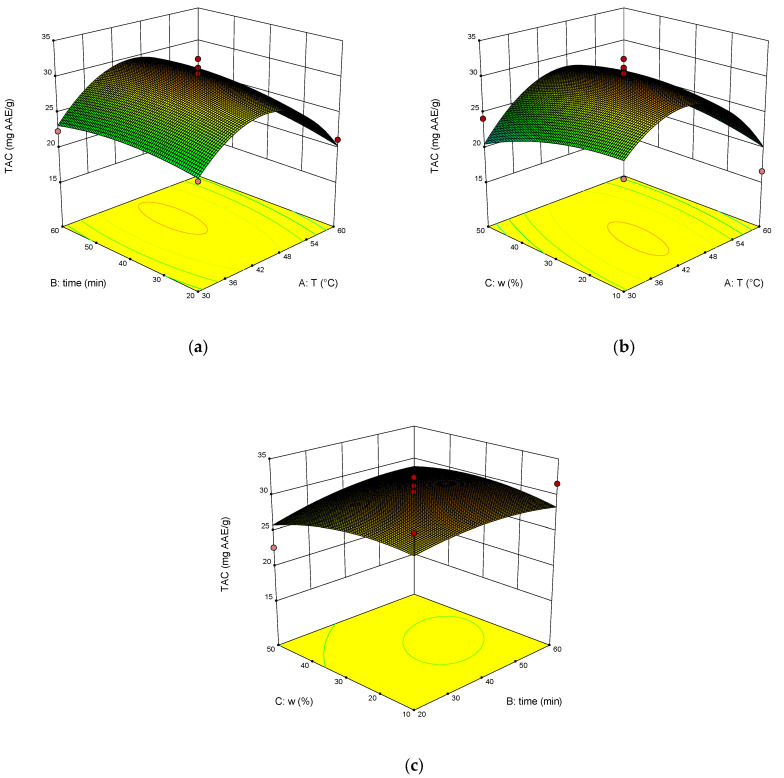
Response surface (**a**–**c**) shows the extraction temperature (A), extraction time (B) and water content (C) effect on the extraction yield of TAC.

**Figure 6 molecules-28-00912-f006:**
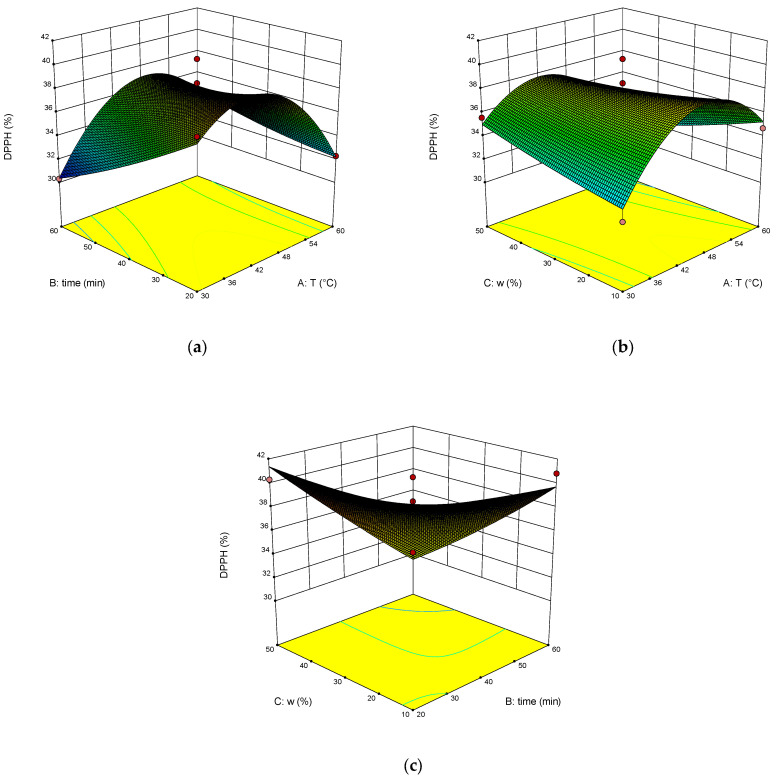
Response surface (**a**–**c**) shows the extraction temperature (A), extraction time (B) and water content (C) effect on the extraction yield of DPPH.

**Figure 7 molecules-28-00912-f007:**
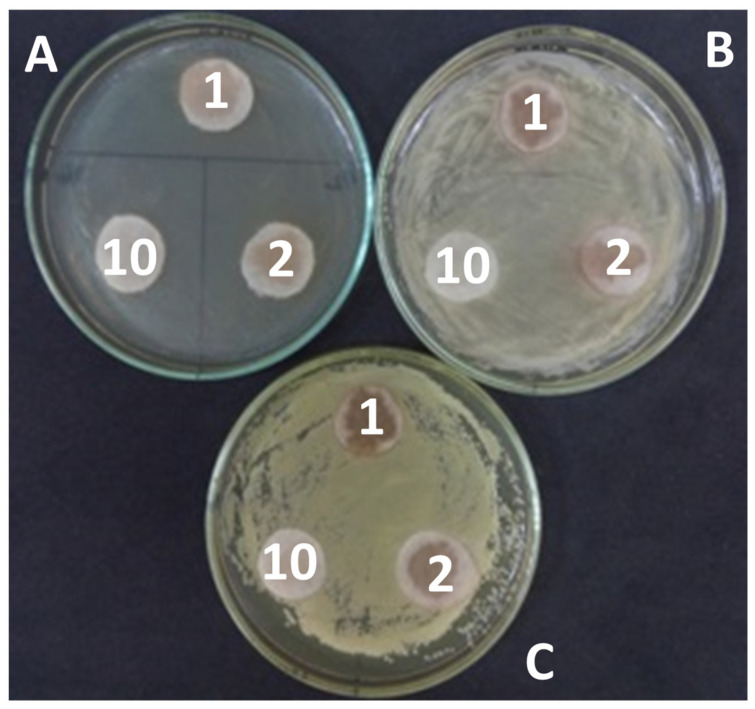
Experiment with bacterial cultures and extract in water at the initial concentration (“1”) and diluted 2 and 10 times (“2” and “10” respectively). (**A**)—*Micrococcus luteus*, (**B**)—*Pseudomonas fluorescens*, (**C**)—*Bacillus subtilis*.

**Figure 8 molecules-28-00912-f008:**
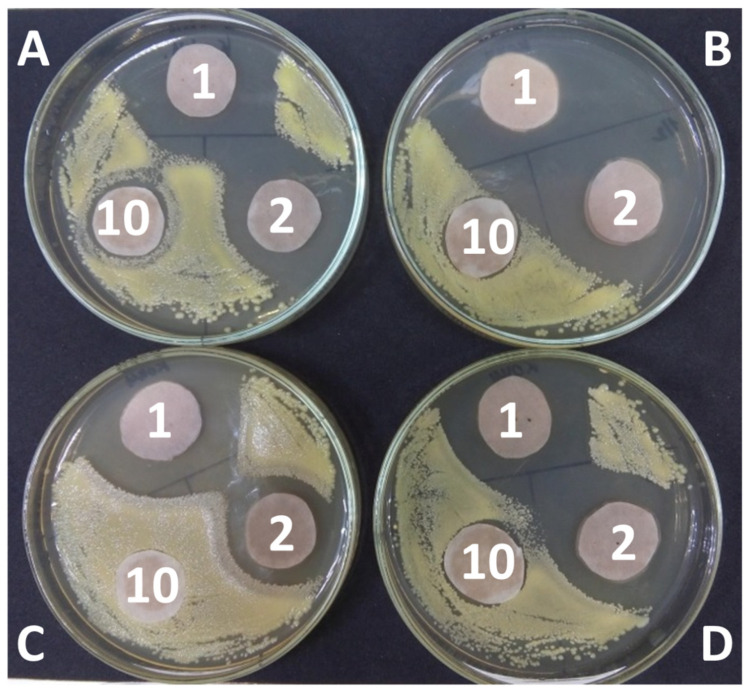
Experiment with *Pseudomonas fluorescens* and extracts at initial concentration and diluted 2 and 10 times. (**A**)—NADES with malonic acid, (**B**)—with malic acid, (**C**)—with tartaric acid, and (**D**)—with citric acid.

**Table 1 molecules-28-00912-t001:** Results of Tukey’s HSD test of estimation a statistically significant difference at *p* ≤ 0.05 between extraction efficiency of various solvents using parameters TPC, TAC and DPPH.

Group 1	Group 2	TPC		TAC		DPPH	
		*p*-Value	Significance	*p*-Value	Significance	*p*-Value	Significance
Water	EtOH	<0.001	Yes	<0.001	Yes	0.3	No
Water	MA	<0.001	Yes	<0.001	Yes	<0.001	Yes
Water	Mal	<0.001	Yes	<0.001	Yes	<0.001	Yes
Water	Tar	<0.001	Yes	<0.001	Yes	0.029	Yes
Water	CA	<0.001	Yes	<0.001	Yes	<0.001	Yes
EtOH	MA	<0.001	Yes	<0.001	Yes	<0.001	Yes
EtOH	Mal	<0.001	Yes	<0.001	Yes	<0.001	Yes
EtOH	Tar	<0.001	Yes	<0.001	Yes	0.556	No
EtOH	CA	0.949	No	<0.001	Yes	<0.001	Yes
MA	Mal	<0.001	Yes	1.000	No	<0.001	Yes
MA	Tar	0.002	Yes	<0.001	Yes	<0.001	Yes
MA	CA	<0.001	Yes	0.999	No	<0.001	Yes
Mal	Tar	<0.001	Yes	<0.001	Yes	0.029	Yes
Mal	CA	<0.001	Yes	0.999	No	0.076	No
Tar	CA	<0.001	Yes	<0.001	Yes	<0.001	Yes

**Table 2 molecules-28-00912-t002:** ANOVA results for optimizing extraction conditions for biologically active substances from *R. rosea* (L.) when NADES choline chloride + malonic acid are used.

Source	Sum of Squares				Mean Square				F-Value				*p*-Value			
	RY	TPC	TAC	DPPH	RY	TPC	TAC	DPPH	RY	TPC	TAC	DPPH	RY	TPC	TAC	DPPH
Model	394,689.3	30,209.9	230.0	152.8	43,854.4	3356.7	25.6	17.0	4.94	4.09	1.93	6.41	0.023	0.038	0.198	0.011
A-T	3702.5	186.2	4.1	0.8	3702.5	186.2	4.1	0.8	0.42	0.23	0.31	0.32	0.539	0.648	0.597	0.590
B-t	107.1	4777.5	4.4	12.8	107.1	4777.5	4.4	12.8	0.01	5.83	0.33	4.81	0.916	0.047	0.584	0.064
C-w	312,504.7	270.3	2.2	1.5	312,504.7	270.3	2.2	1.5	35.21	0.33	0.17	0.58	0.001	0.584	0.695	0.472
AB	85.1	2337.7	2.4	22.1	85.1	2337.7	2.4	22.1	0.01	2.85	0.18	8.34	0.925	0.135	0.683	0.023
AC	415.2	18.9	15.2	10.9	415.2	18.9	15.2	10.9	0.05	0.02	1.15	4.11	0.835	0.884	0.319	0.082
BC	8713.8	475.2	2.3	19.8	8713.8	475.2	2.3	19.8	0.98	0.58	0.17	7.47	0.355	0.471	0.692	0.029
A^2^	14,021.1	14,466.2	180.5	84.8	14,021.1	14,466.2	180.5	84.8	1.58	17.64	13.66	32.00	0.249	0.004	0.008	0.001
B^2^	32,249.5	620.5	4.2	0.5	32,249.5	620.5	4.2	0.5	3.63	0.76	0.32	0.18	0.098	0.413	0.591	0.683
C^2^	23,800.1	5408.4	6.3	0.1	23,800.1	5408.4	6.3	0.1	2.68	6.60	0.48	0.06	0.146	0.037	0.512	0.820
Residual	62,123.6	5739.6	92.5	18.5	8874.8	819.9	13.2	2.6								
Lack of Fit	15,010.7	5584.1	74.4	6.0	5003.6	1861.4	24.8	2.0	0.42	47.86	5.49	0.64	0.746	0.001	0.067	0.627
Pure Error	47,112.9	155.6	18.1	12.5	11,778.2	38.9	4.5	3.1								
Cor Total	456,812.8	35,949.5	322.5	171.3												

**Table 3 molecules-28-00912-t003:** The number of bacterial cells in solution (c/mL) when extracts are used.

Solvent	*Escherichia coli* (10,000 c/mL) *			*Pseudomonas* sp. (15,000 c/mL) *			*Micrococcus* sp. (15,000 c/mL) *		
	Extracts Concentration (%)								
	2	1	0.5	2	1	0.5	2	1	0.5
Tar	30 ± 6	100 ± 23	7400 ± 760	0	70 ± 9	100 ± 15	0	0	0
CA	30 ± 10	120 ± 31	6700 ± 920	0	0	0	0	0	0
MA	0	0.2 ± 0.009	4200 ± 230	0	0	0	0	0	0
Mal	0	0.15 ± 0.01	7000 ± 870	0	0	0	0	0	0

* Initial number of bacteria.

**Table 4 molecules-28-00912-t004:** The number of bacterial cells of *Bacillus brevis* in solution (c/mL) when different extracts (initial number, 20,000 c/mL) are used.

Solvent	Extract Concentration (%)	
	50	2
Tar	10,100 ± 1090	18,000 ± 2150
CA	5080 ± 890	19,500 ± 3060
MA	1000 ± 58	19,600 ± 2790
Mal	14,800 ± 1100	19,000 ± 3120

**Table 5 molecules-28-00912-t005:** The number of bacterial cells in solution (c/mL) when an aqueous extract is used.

Strain	Initial	Extract Concentration (%)		
	Number (c/mL)	50	2	1
*Escherichia coli*	10,000 ± 890	3500 ± 270	936 ± 60	9290 ± 1060
*Pseudomonas* sp.	25,000 ± 2150	15 ± 2	60 ± 5	130 ± 10
*Bacillus brevis*	15,000 ± 1050	14,600 ± 1500	14,800 ± 2800	13,700 ± 1200
*Micrococcus* sp.	20,000 ± 1870	85 ± 17	12,000 ± 2460	16,600 ± 1770

**Table 6 molecules-28-00912-t006:** The number of *Escherichia coli* bacterial cells in solution (c/mL) when extracts and solvents at a concentration of 5% (initial number, 65,000 ± 5430 c/mL) are used.

	Extract	Solvent
Tar	890 ± 60	10,280 ± 880
CA	210 ± 13	12,000 ± 1085
MA	0	30 ± 2
Mal	0	0

## Data Availability

Not applicable.

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
