# Peer review of "Application of Natural Deep Eutectic Solvents for Extraction of Bioactive Components from Rhodiola rosea (L.)"

_molecules, 2023, doi:10.3390/molecules28020912_

Round 1
Reviewer 1 Report
Please check:
line 68:Choline chloride
Line 103 and 104: Folin Ciocalteu
Fig2 chart description please add Temperature, time used
Author Response
Dear Reviewer,
Thank you for your comments and suggestion!
Comment: line 68:Choline chloride
Reply: Necessary changes have been made to the text of the article.
Results and Discussion Line 103 and 104: Folin Ciocalteu
Reply: Necessary changes have been made to the text of the article.
Results and Discussion Fig2 chart description please add Temperature, time used
Reply: Necessary changes have been made to the text of the article.
Reviewer 2 Report
In general, paper is nicely written and the idea behind the research is highly interesting. The use of NADES is still developing field, and bioactive components from medicinal plants are of high value for further research.
Some of the suggestions for authors:
1. It is highly suggested that you change the term "useful substances" for some more appropriate- I am suggesting active principles, active compounds... any other term sounds much better.
2. But this plant is rare in wild nature of some regions.- Please rephrase the sentence to The occurrence of this plant is rare in wild nature of some regions. More frequently, the plant is cultivated; however, active compounds are accumulated....
3. Also in the section introduction- bearing in mind that antibacterial activity is chosen for testing, please add some data of traditional medicine, whether this plant is used traditionally for this purpose.
4. Both Latin name of plant and bacterial strains should be in italic throughout the entire text, please make sure that this is the case.
5. Many of abbreviations are not defined at first appearance- please, make sure that this is corrected throughout the entire text.
6. Also, many sentences are starting with abbreviation- this is generally not recommended and it be should changed.
7. Sections 2.2.2 and 2.2.3 should be reorganized. Investigation of TPC is generally considered as preliminary chemical characterization and TAC and DPPH are biological activities. Section 2.2.3 should be renamed to Analysis of rosavin by HPLC
8. In section 2.2.4, the use of active should be changed to passive, from "We used" to "....was used".
9. The correct name is Petri dish.
10. In general, section 2.2.4 should be expanded. How was the identity of bacteria confirmed? How were the strains cultivated, for how long? How was the density of bacterial suspension determined? Which density was used? How long were the inoculated Petri dishes cultivated? Etc. Please add more data, and possibly separate the disk diffusion test from other experiments, each giving enough details for understanding. Additional question-was there any positive control, disk impregnated with some antibiotic?
11. Section Results and Discussion is generally written nicely. However, the lack of references towards other papers investigating at least similar matter is a serious flaw of this paper. Authors have provided some references in Introduction, suggesting that there were other papers on these matter- please, add this segment of Discussion, as this section is generally devoted to comparison of obtained results with previously published material. If there are no other papers investigating NADES on R.rosea, please add some comment and data on conventional methods of extractions in this plant.
12. When results of antibacterial testing are given, it is more frequently reported as susceptibility of microbial strain, rather than giving expression "resistant". Please make the necessary changes.
13. Regarding Tables 3-5- why did the authors choose precisely these initial concentrations of bacterial? Why were they not the same for all strains?
14. In same tables, please add the percentage of decrease of bacteria number, as it is sometimes easier to make conclusions on this matter.
15. Section Conclusions should be re-written, and leaving only the most important numbers. At the current state, Conclusion looks more like another Abstract.
Author Response
Dear Reviewer,
Thank you for your comments and suggestion! They allowed us to make our manuscript much better. We have taken into account all your comments and made the necessary corrections.
Comment: It is highly suggested that you change the term "useful substances" for some more appropriate- I am suggesting active principles, active compounds... any other term sounds much better.
Reply: Thank you for your comment. We changed this term.
- But this plant is rare in wild nature of some regions.- Please rephrase the sentence to The occurrence of this plant is rare in wild nature of some regions. More frequently, the plant is cultivated; however, active compounds are accumulated....
Reply: Thank you for your comment. We changed this phrase
Comment: Also in the section introduction- bearing in mind that antibacterial activity is chosen for testing, please add some data of traditional medicine, whether this plant is used traditionally for this purpose.
Reply: Thank you for your suggestion. We added more information into Introduction.
Comment: Both Latin name of plant and bacterial strains should be in italic throughout the entire text, please make sure that this is the case.
Reply: Latin names have been checked. Necessary changes have been made to the text of the article.
Comment: Many of abbreviations are not defined at first appearance- please, make sure that this is corrected throughout the entire text.
Reply: Necessary changes have been made to the text of the article.
Comment: Also, many sentences are starting with abbreviation- this is generally not recommended and it be should changed.
Reply: Thank you for your comment. Necessary changes have been made to the text of the article.
Comment: Sections 2.2.2 and 2.2.3 should be reorganized. Investigation of TPC is generally considered as preliminary chemical characterization and TAC and DPPH are biological activities. Section 2.2.3 should be renamed to Analysis of rosavin by HPLC
Reply: Sections have been reorganized.
Comment: In section 2.2.4, the use of active should be changed to passive, from "We used" to "....was used".
Reply: Thank you for your comment. We changed active to passive form.
Comment: The correct name is Petri dish.
Reply: Necessary changes have been made to the text of the article.
Comment: In general, section 2.2.4 should be expanded. How was the identity of bacteria confirmed? How were the strains cultivated, for how long? How was the density of bacterial suspension determined? Which density was used? How long were the inoculated Petri dishes cultivated? Etc. Please add more data, and possibly separate the disk diffusion test from other experiments, each giving enough details for understanding. Additional question-was there any positive control, disk impregnated with some antibiotic?
Reply: The studied bacterial strains were obtained from the All-Russian Collection of Microorganisms - VKM. The obtained strains were cultivated on meat-peptone agar in the Museum of bacteria and fungi of Kola peninsula INEP Herbarium of Institute of the Industrial Ecology Problems of the North of the Kola Science Center of the Russian Academy of Sciences with periodic subculture for several years.
The necessary additions were made to the text of the article.
Comment: Section Results and Discussion is generally written nicely. However, the lack of references towards other papers investigating at least similar matter is a serious flaw of this paper. Authors have provided some references in Introduction, suggesting that there were other papers on these matter- please, add this segment of Discussion, as this section is generally devoted to comparison of obtained results with previously published material. If there are no other papers investigating NADES on R.rosea, please add some comment and data on conventional methods of extractions in this plant.
Reply: Thank you for your suggestion. We added more information into Results and Discussion.
Comment: When results of antibacterial testing are given, it is more frequently reported as susceptibility of microbial strain, rather than giving expression "resistant". Please make the necessary changes.
Reply: Necessary changes have been made to the text of the article.
Comment: Regarding Tables 3-5- why did the authors choose precisely these initial concentrations of bacterial? Why were they not the same for all strains?
Reply: The studied number of bacteria in body fluids, such as urine, is considered to be quite high and may indicate the presence of an inflammatory process. The fluctuation of the number within the order in different variants of experiments is not significant.
Comment: In same tables, please add the percentage of decrease of bacteria number, as it is sometimes easier to make conclusions on this matter.
Reply: Тhe percentage of reduction in the bacteria number has been calculated, but it does not reflect the standard deviation and distorts the data.
Comment: Section Conclusions should be re-written, and leaving only the most important numbers. At the current state, Conclusion looks more like another Abstract.
Reply: Thank you for your suggestion. Conclusion has been re-written.
Round 2
Reviewer 2 Report
The authors have made the necessary changes.
Author Response
Thank you very much.